# High-Speed 600 GHz-Band Terahertz Imaging Scanner System with Enhanced Focal Depth

**Yaheng Wang [1], Li Yi [1], Masayoshi Tonouchi [2] and Tadao Nagatsuma [1,\*]**

[1] Graduate School of Engineering Science, Osaka University, Osaka 560-8531, Japan
[2] Institute of Laser Engineering, Osaka University, Osaka 565-0871, Japan
\* Correspondence: nagatuma@ee.es.osaka-u.ac.jp; Tel.: +81-06-6850-6336

**Abstract:** Lenses/mirrors with fast data acquisition speeds and extended focal depths have practical importance in terahertz (THz) imaging systems. Thus, a high-speed 600 GHz-band THz imaging scanner system with enhanced focal depth is presented in this work. A polygon mirror with a 250 Hz scanning frequency and an integrated off-axis parabolic (OAP) mirror with an ~170 mm focal depth were employed for 2D imaging. The simulation and experimental results show that a spatial resolution of ~2 mm can be achieved as the imaging distance varies from ~85 to 255 mm. The proposed system was applied to image a hidden metal object as a potential security application, demonstrating that this system can image targets with an enhanced focal depth.

**Keywords:** polygon mirror; integrated off-axis parabolic mirror; enhanced focal depth; terahertz imaging

## 1. Introduction

In recent years, terahertz (THz) wave imaging systems have attracted considerable attention due to their millimeter and submillimeter resolution and transparency to electrical insulators such as paper, plastics, and glass. THz imaging techniques are, therefore, expected to be used in various applications, such as security, nondestructive inspection, and noncontact inspection [1–3].

Objects with fast moving speeds and uncertain positions are common in practical applications, and THz arrays may be a viable solution that satisfies both of these requirements. However, this technology possesses various technical difficulties and prohibitive costs [4]. In general, beam focusing with quasi-optical components, combined with mechanical beam scanning systems, can be applied to 2D imaging with sufficient spatial sampling, which has the advantages of fast data acquisition speed and cost-effectiveness [5–7]. However, the imaging distance is fixed according to the focal length of the lenses/mirrors, which limits the imaging distance of the targets. Thus, to apply THz imaging systems in industrial applications, the imaging speed, imaging aperture, and flexible imaging distances should be considered.

Some related studies on THz imaging systems are summarized in Table 1. A quasi-optical scanning configuration is a cost-effective approach for achieving fast imaging speeds when the array technique cannot be applied. In [4–8], scanning mirrors, such as galvanometer mirrors and polygon mirrors, were installed to improve the imaging speed. Previously, we presented a 300 GHz-band THz imaging system with a 40 Hz galvanometer mirror that achieved an imaging speed of 80 mm/s [4]. Similarly, a 210 GHz THz imaging system with an 80 Hz polygon mirror achieved an imaging speed of 250 mm/s [9]. More recently, a 410 GHz-band imaging system with a 200 Hz polygon mirror was installed to obtain an imaging speed of 40 mm/s [10]. Notably, none of these systems considered the focal depth. Thus, the 2D imaging was limited to fixed imaging distances.

Several methods can be used to enhance the focal depth, such as parallel plate waveguides [11], quasi-Bessel beams [12], and meta-surfaces [13]. These techniques effectively

increase the focal depth. For example, diffraction-free quasi-Bessel beams have been used in 300 GHz-band systems to achieve an enhanced focal depth of 155 mm [12]. However, the imaging speed was not considered in these systems, and slow imaging speeds with two-axes moving stages are not practical. Moreover, these methods required additional components that increase the cost and complexity of the system. In comparison, an off-axis parabolic (OAP) mirror possesses several advantages, such as no central obscuration, a wide field of view, and an enhanced focal depth [14]. Remarkably, the astigmatic aberration of the OAP mirror increases the focal depth but reduces the spatial resolution to some extent. In this work, an integrated OAP mirror was designed based on the previous concave mirror and the high-speed 250 Hz polygon mirror [4]. The proposed system aims to achieve better performance concerning high-speed imaging, more extended focal depth, and better resolution, as shown in Table 1.

**Table 1.** Comparison of this work with other THz imaging systems.

| Parameter | This Work | Ref [9] | Ref [10] | Ref [12] |
|---|---|---|---|---|
| Frequency | 600 GHz | 210 GHz | 410 GHz | 300 GHz |
| Scanning frequency | 250 Hz | 80 Hz | 200 Hz | / |
| Stage speed | 500 mm/s | 250 mm/s | 40 mm/s | / |
| Spatial resolution | 2 mm | 2.83 mm | 1.26mm | 3 mm |
| Focal depth | 170 mm | 8 mm | 7 mm | 155 mm |

The remainder of this paper is organized as follows. The system configuration and evaluation of the high-speed 600 GHz-band THz imaging system using a polygon mirror is introduced in Section 2. To improve the focal depth of the scanner lens/mirror, an integrated OAP mirror was proposed. The design and evaluation of the OAP mirror, as well as the numerical simulation results, are discussed in Section 3. Moreover, the proposed OAP mirror was installed in the high-speed imaging system, and the performance was experimentally evaluated. Finally, the conclusions and directions for future work are described in Section 4.

## 2. Polygon Mirror for 600 GHz-Band Terahertz High-Speed Imaging

### 2.1. Approach for Achieving High Acquisition Speed

In general, beam focusing systems with quasi-optical components or synthetic aperture radar (SAR) can be applied to 2D and 3D imaging [15–17]. Quasi-optical systems provide high resolution defined by the diffraction limit [18], whereas the imaging distance is fixed according to the focal length of the lens. Therefore, combining quasi-optical systems with mechanical beam scanning systems to obtain faster acquisition speeds in 2D imaging is a cost-effective method [10–19].

In our previous THz imaging systems, focusing elements, such as f-theta lenses or concave mirrors, were used in combination with galvanometer mirrors for 2D imaging [4]. The imaging speed along the moving direction was limited to 100 mm/s due to the limited rotating speed of the galvanometer mirror. By introducing a polygon mirror, the scanning speed can be improved [6,8–10]. In this case, we introduced a polygon mirror with a scanning speed of 250 Hz to our THz imaging system.

A mechanical beam scanning system using a polygon mirror and an f-theta lens for 2D imaging is shown in Figure 1a. The incident collimated beam is reflected by a polygon mirror and focused by an f-theta lens. The focal length of the f-theta lens is 160 mm, and the distance between the polygon mirror and the f-theta lens is 100 mm. Figure 1b shows some of the 1D imaging data collected using the polygon mirror, and the effective pixel number in the scanning direction is one-third of the total number of measurement points. The experimental results show that ~30% of the beam is scanned across the f-theta lens, while the remainder of the pixels are wasted. A larger scannable mirror can be introduced to improve the efficiency of the system.

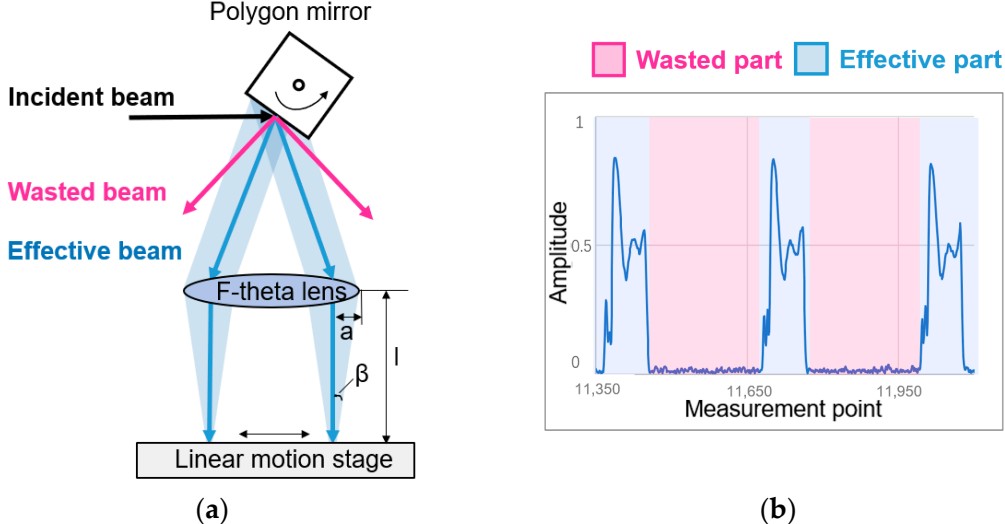

**Figure 1.** (**a**) Schematic diagram of the beam reflected by the polygon mirror. (**b**) Part of the 1D imaging data was collected using the polygon mirror.

According to the Rayleigh criterion, the resolution, $r_R$, is related to the wavelength of the beam, $\lambda$, and half of the aperture angle, $\beta$, by the following equation [20]:

$$r_R = \frac{0.61\lambda}{sin\beta} \, , \tag{1}$$

where is the focal length of the f-theta lens and *a* is half of the beam aperture, as shown in Figure 1a. To improve the resolution, a beam with a smaller wavelength should be used. Thus, we used a 600 GHz-band beam, which corresponds to a wavelength of 0.5 mm, as the source in our system. Hence, according to (1), the resolution of this system can be theoretically obtained as 2.44 mm.

### 2.2. Configuration of the High-Speed THz Imaging System

A block diagram of the high-speed THz imaging system and the actual configuration of the developed system are shown in Figures 2a and 3, respectively. A 600 GHz-band THz wave is generated by using a millimeter-wave synthesizer and an 18-times frequency multiplier, and the output power is ~10 microwatts. The THz wave is emitted from the horn antenna and reflected by the parabolic mirror to collimate the beam. Then, the collimated THz beam passes through a 50/50 beam splitter and is reflected by a four-sided polygon mirror. The scanning frequency of the polygon mirror is 250 Hz, and the mirror is triggered with a function generator. The beam reflected by the polygon mirror is then focused by the f-theta lens to scan the beam along the x-direction, that is, the scanning direction of the polygon mirror. Next, the beam is focused onto the surface of a linear motion stage to image the tested sample; the direction of the moving stage is defined as the y-direction. Based on the experimental evaluation and hardware limitation [21], the limitation of our imaging system is: the maximum scanning frequency is 250 Hz and the maximum speed of the moving stage is 500 mm/s.

For the receiver part, the beam reflected by the tested sample is focused by another parabolic mirror, and then detected by a Schottky barrier diode (SBD). The detected signal is amplified with a low-noise amplifier and sampled by a digital multimeter (DMM). Moreover, the THz wave is modulated at 100 kHz and a digital lock-in amplifier algorithm is used to remove the minor reflected signals in the 600 GHz band [22]. In this study, the time constant is 1 μs due to the digital lock-in amplifier algorithm utilized, while the dynamic range decreases from ~30 to ~22 dB. Due to the short imaging distance, the influence of humidity on the experiment is negligible.

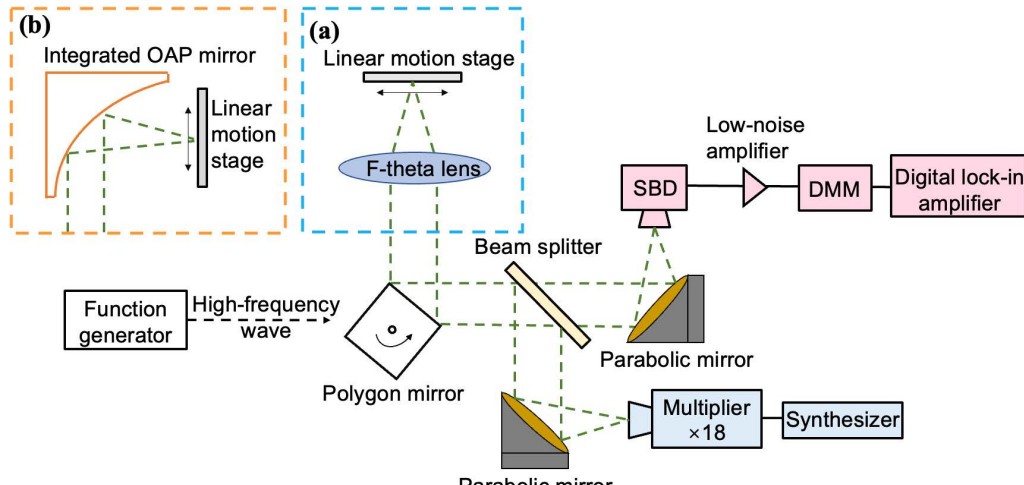

**Figure 2.** Block diagram of the imaging system. (**a**) Focusing element used in high-speed imaging system. (**b**) Focusing element used in the imaging system with enhanced focal depth. (SBD: Schottky barrier diode. DMM: digital multimeter.).

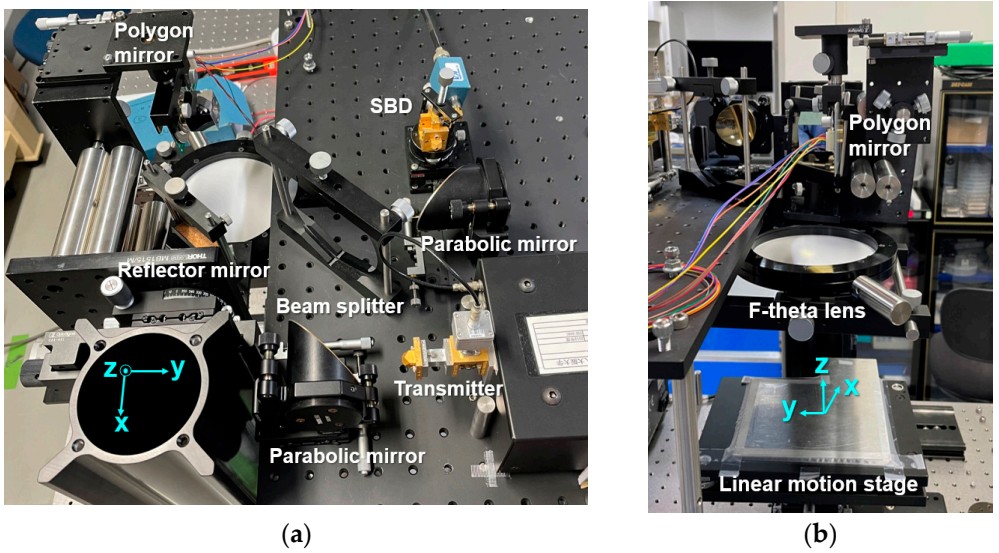

**Figure 3.** Experimental setup of the high-speed imaging system. (**a**) Vertical view of the experimental setup. (SBD: Schottky barrier diode.) (**b**) Overview of the experimental setup.

### 2.3. Imaging Results with a Polygon Mirror

We used a metal erasing shield as an imaging sample, and a photo of the metal erasing shield is shown in Figure 4a. The thinnest metal hole in this sample is 2 mm. According to the imaging result shown in Figure 4b, the resolution limit for distinguishing the 2 mm metal hole could be obtained under the maximum scanning frequency of the polygon mirror, which is 250 Hz, and the maximum speed of the moving stage, which is 500 mm/s. Moreover, to image objects with not flat surfaces, the hexagon wrench set was used as an imaging sample as shown in Figure 5a at the same imaging speed; the size of the hexagon wrench was 1.5, 2.0, 2.5, 3.0, and 5.0 mm, respectively. The hexagon wrench set was placed on the moving stage with a metal plate. Figure 5b,c show the imaging results of the hexagon wrench set in vertical and horizontal directions; the thinnest 1.5 mm hexagon wrench can be clearly distinguished. We can image a $120 \times 500 \text{ mm}^2$ area in one second with the proposed system. Notably, the imaging area per second is ~6.7 times higher than our previous imaging system.

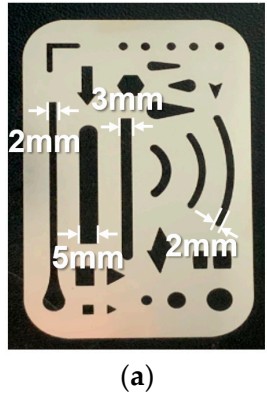
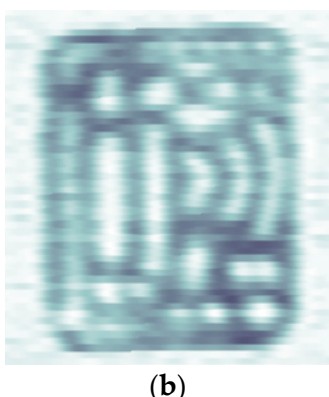

(**a**)

(**b**)

**Figure 4.** (**a**) Photo of the metal erasing shield. (**b**) The imaging result of the metal erasing shield.

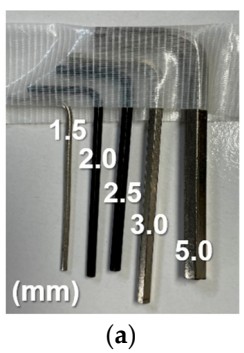
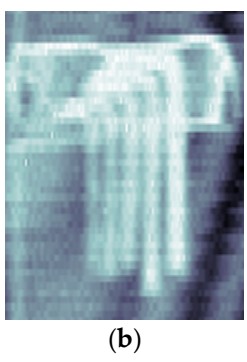
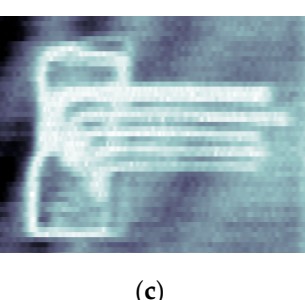

(**a**)

(**b**)

(**c**)

**Figure 5.** (**a**) Photo of the hexagon wrench set. (**b**) The imaging result of the hexagon wrench set in the vertical direction. (**c**) The imaging result of the hexagon wrench set in the horizontal direction.

However, the result is worse than that reported in [23] because a scannable lens, such as an f-theta lens, reduces the spatial resolution of the system. There are two main reasons for this result: first, the f-theta lens was not specifically designed for the 600 GHz band but was simulated according to certain assumptions about the optical domain; second, the optical alignment is very sensitive, and the resolution must be reduced to obtain the appropriate imaging aperture. In addition, together with the beam size issue mentioned in Section 2.1, the design of the scannable lens should be improved further for use in the 600 GHz band.

## 3. 600 GHz-Band Terahertz Imaging Scanner System with Enhanced Focal Depth

### 3.1. Approach for Achieving a Longer Focal Depth

In Section 2, we used a mechanical beam-scanning system with a polygon mirror and an f-theta lens for 2D imaging. However, the f-theta lens has a limited focal depth of ~5 mm, and this insufficient focal depth only supports 2D imaging at fixed imaging distances. As previously discussed, there are many methods for enhancing the focal depth, but additional components increase the complexity and costs of the system. It is mentioned in [14,24–26] that the OAP mirror exhibits an optical phenomenon called astigmatism, which can enhance the focal depth. Therefore, we aimed to utilize this phenomenon to design a focusing mirror with a longer focal depth.

The parabolic mirror is a conventional optical component that can focus collimated beams in imaging applications. When the beam enters along the optical axis, the ideal spatial resolution can be obtained, and the focal depth, $d_e$, can be calculated [27] as:

$$d_e = \frac{2\pi\omega_0^2}{\lambda} \,, \tag{2}$$

where $\omega_0$ is the beam waist and $\lambda$ is the wavelength of the beam. For the THz source operated at 600 GHz, the focal depth was approximately 10 mm, which is usually insufficient for detecting objects with uncertain positions.

However, the focal depth of a parabolic mirror increases when the incident beam deviates from the optical axis. This phenomenon refers to a monochromatic aberration, namely, astigmatism, caused by narrow incident beams with large inclination angles that deviate from the optical axis. The horizontal and vertical beams are focused at different focal points, as shown in Figure 6 [28–30]. Although no quantitative evaluations of the relationship between the increased focal depth and incident angle have been detailed for the OAP mirror, a comprehensive study was presented for the spherical mirror, as reported in [28]. When the beam is incident at an off-axis angle, $\gamma$, the focal depth, $d_a$, can be determined as:

$$d_a = \frac{rf}{2f\cos\gamma - r} - \frac{rf\cos\gamma}{2f - r\cos\gamma} ,\qquad(3)$$

where $r$ is the radius and $f$ is the focal length. When the beam is incident at a larger off-axis angle from the optical axis, the focal depth increases, and the resolution decreases. Small OAP mirrors intercept only small parts of the parabolic mirror. Due to their small size, we approximated small OAP mirrors as spherical mirrors. The parameters of the center small OAP mirror were defined as follows: the radius was 180 mm, the focal length was 100 mm, the off-axis angle was 5 degrees, and the focal depth was approximately 68 mm. Thus, when a narrow THz beam is incident on a parabolic mirror with an off-axis angle, a longer focal depth than that defined in (2) can be obtained.

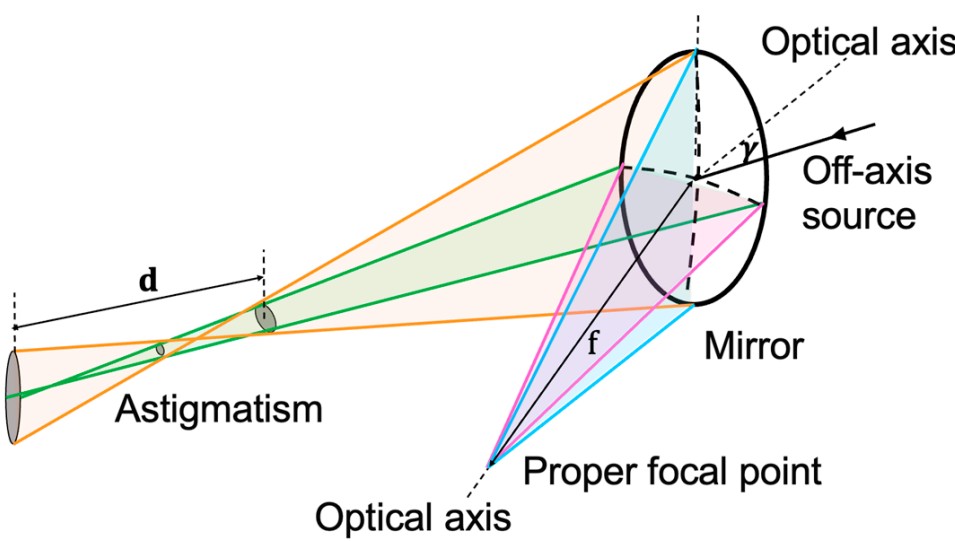

**Figure 6.** Schematic of astigmatism with the spherical mirror.

### 3.2. Design of the Integrated OAP Mirror

Based on a previous design of a concave mirror [4], an integrated OAP mirror is proposed in this study, as shown in Figure 7. Three small OAP mirrors are assembled as an integrated OAP mirror, which covers the required imaging area. The collimated beam is reflected by a rotating polygon mirror onto the integrated OAP mirror and focused on the imaging plane. The design of the proposed mirror is divided into three procedures: the design of the base OAP mirror, the design of each small OAP mirror, and the coordinate translation of each small OAP mirror.

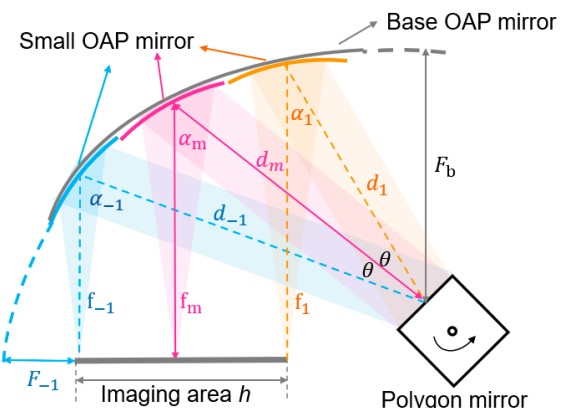

**Figure 7.** Schematic of the integrated OAP mirror, which includes three small OAP mirrors; the dashed line indicates the base OAP mirror, which guides the parameter design of each small OAP mirror.

First, the base OAP mirror is defined to achieve a 1D scanning range as follows:

$$z = \frac{(x^2 + y^2)}{4F_b} - F_b, \tag{4}$$

where $x$, $y$, and $z$ denote the surface coordinates and $F_b$ is the parent focal length of the base OAP mirror. Based on the practical considerations of our imaging scanner system, the imaging area, $h$, is determined to be 150 mm, and the rotation angle of the polygon mirror, $\theta$, is set to 30 degrees. To calculate the other parameters of the base OAP mirror and small OAP mirrors, the focal length, $f_m$, and off-axis angle, $\alpha_m$, of the center OAP mirror are defined as 100 mm and 50 degrees, respectively. Then, $F_b$ can be calculated as:

$$F_b = \frac{h}{8} \cdot \cot \theta \cdot (1 + \cos \alpha_m) - \frac{\sin \alpha_m}{4} . \tag{5}$$

Next, after the base OAP mirror is fixed, the parameters of each small OAP mirror can be determined as follows. The off-axis angle of each small OAP mirror $\alpha_i$ ($i = \pm 1, 2 \ldots$ n) can be deduced according to $\alpha_m$ and $\theta$ with the following geometrical relation:

$$\theta = \alpha_{-1} - \alpha_m = \alpha_m - \alpha_1 . \tag{6}$$

Then, the distance between the focal point of the base OAP mirror and the center of each small OAP mirror, which is defined as $d_i$, can be calculated as:

$$d_i = F_b \left( 1 + \frac{2(1 - \cos \alpha_i)}{\sin \alpha_i} \right). \tag{7}$$

In addition, the focal point of the base OAP mirror in the imaging plane is constant, which can be expressed as:

$$d_1 + f_1 = d_m + f_m = d_{-1} + f_{-1}. \tag{8}$$

As $f_m$ was calculated to be 100 mm, the focal lengths of each small OAP mirror $f_i$ can be determined. Hence, the parent focal length of each small OAP mirror $F_i$ can be calculated using:

$$F_i = \frac{f_i \cdot \sin^2 \alpha_i}{\sin^2 \alpha_i + (1 - \cos^2 \alpha_i)} . \tag{9}$$

In this case, the equations of all of the small OAP mirrors can be determined. Finally, the integrated OAP mirror can be assembled by placing each small OAP mirror at its center coordinate and rotating it by $\alpha_i$. As the OAP mirrors cannot connect perfectly due to the

limited number of mirrors used, numerical interpolation was applied to fill in the gaps between the mirrors.

Two main factors affect the resolution of the integrated OAP mirror: the focal length and the number of OAP mirror segments. The relation between these two factors and the resolution was quantitatively examined using numerical simulations. The results indicate that a smaller focal length leads to a decreased resolution, whereas more OAP segments lead to a more stable and higher resolution. According to the imaging results and the fabrication difficulty, we used seven OAP segments in our system. Based on the parameters obtained with the above calculations, one of the numerical simulation results with the proposed integrated OAP mirror is shown in Figure 8a. Figure 8b shows the simulation results obtained on the imaging plane when the rotation angle of the polygon mirror is 0, 5, 10, 20, and 25 degrees.

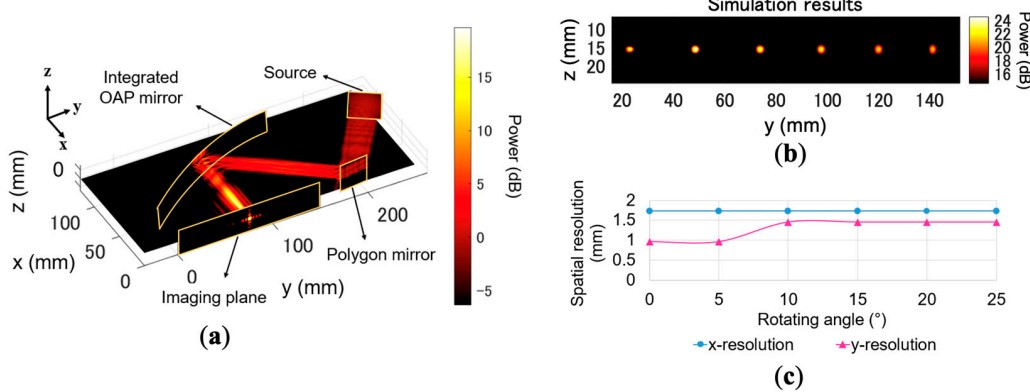

**Figure 8.** (**a**) Simulation results of the integrated OAP mirror beam path with the proposed setup. (**b**) Simulation results of the imaging plane at different incident angles. (**c**) Simulation results of the spatial resolution of both axes. The blue dots represent the resolution in the y-direction, and the pink triangles indicate the resolution in the z-direction.

The spatial resolution was evaluated using the 10/90 knife-edge method [31]. For the ideal focal length of 100 mm, Figure 8c shows the spatial resolution obtained by the simulation, with resolutions of ~1.73 and ~1.45 mm in the y- and z-directions, respectively. However, due to the calculation cost, the complicated optical structure of the proposed integrated OAP mirror, and the differences between the optical and THz domains, the resolution of each pixel and focal depth were not evaluated quantitatively. The spatial resolution and focal depth of the proposed integrated OAP mirror were evaluated further using experiments.

### 3.3. Imaging Experiment with Integrated OAP Mirror

A block diagram of the imaging scanner system and the actual configuration of the proposed system are shown in Figures 2b and 9, respectively. Different from the experimental setup in Section 2.2, the focusing element is replaced by the f-theta lens (Figure 2a) with the integrated OAP mirror (Figure 2b). The integrated OAP mirror is made of aluminum and coated with gold. The beam reflected by the polygon mirror was focused using the integrated OAP mirror to scan the beam along the y-direction. Then, the beam was focused onto the surface of a linear motion stage to image the tested object. The motion direction of the stage was defined as the z-direction. For the receiver part, it is consistent with Section 2.2.

Due to the astigmatism of the OAP mirror, the horizontal and vertical beams were focused on different focal points. In the experiment, we changed the object distance, d, which is defined as the distance between the moving stage and the integrated OAP mirror, to obtain the imaging result. The focal depth range of the integrated OAP mirror was determined by evaluating the imaging results. The scanning frequency of the polygon

mirror was 80 Hz, and the moving stage was moved along the z-direction at a speed of 20 mm/s.

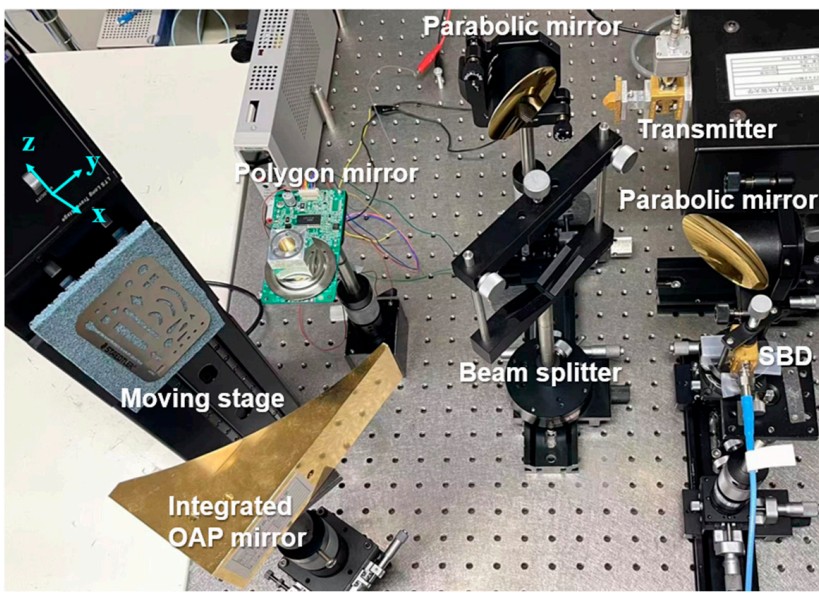

**Figure 9.** Experimental setup of the imaging scanner system. (SBD: Schottky barrier diode.)

We used a metal erasing shield as an imaging object, as shown in the photograph in Figure 10a. The smallest metal hole in the object has a width of 2 mm. Figure 10b shows the imaging results of the metal erasing shield at the objective distance of 13.5 cm; the thinnest 2 mm metal hole can be distinguished. To measure the resolution under different object distances, we moved the stage away from the integrated OAP mirror and imaged the metal erasing shield sample at 2.5 cm intervals. For a better comparison, we measured the 1D data of the middle part of the object, as shown by the orange square in Figure 10a. From top to bottom, Figure 10c shows the 1D imaging results at object distances of 8.5, 11, 13.5, 16.0, 18.5, 21.0, 23.5, and 25.5 cm. The signal quality does not change significantly at the object distance from 8.5 cm to 25.5 cm. When the object distance exceeds 25.5 cm, the focusing performance of the integrated OAP mirror becomes worse, and the signal power received becomes weaker. Therefore, the maximum object distance considered in this experiment is 25.5 cm. The 2D imaging results at different object distances can be found in [32]. The imaging results indicate that the 2 mm metal hole could be distinguished at object distances between 8.5 and 25.5 cm.

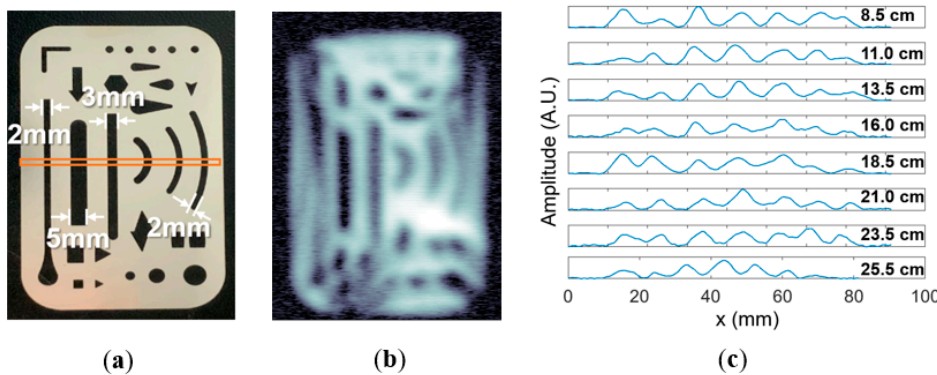

**Figure 10.** Imaging results using the proposed imaging scanner system. (**a**) Photograph of the metal erasing shield. (**b**) Imaging results of the complete metal erasing shield at the objective distance of 13.5 cm. (**c**) Imaging results of a partial metal erasing shield at different object distances.

Moreover, in order to demonstrate the focal depth of the integrated OAP mirror intuitively, we used a layered metal object that consists of a square metal spacer and an L-shaped metal connector as an imaging object, as shown in Figure 11a. The distance from the front surface of the metal spacer to the metal connector is 70 mm, and the object distance from the integrated OAP mirror to the front surface of the metal spacer is 10 cm. The scanning frequency of the polygon mirror was 80 Hz, and the layered metal object on the moving stage was moved along the z-direction at a speed of 20 mm/s. Figure 11c shows the imaging result of the layered metal object; the square metal spacer and metal connector that are 70 mm apart can be identified.

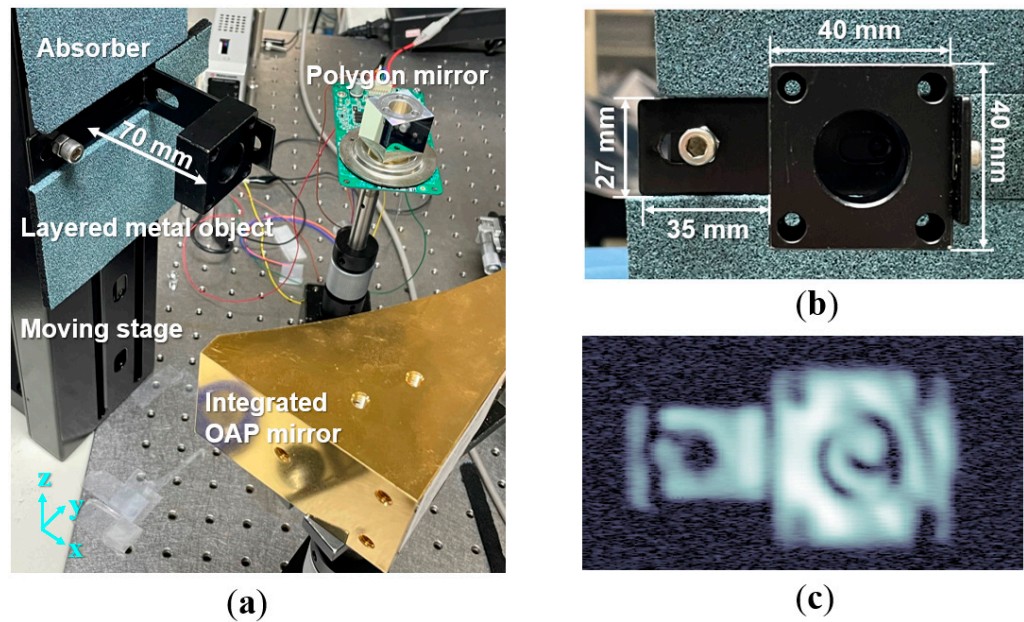

**Figure 11.** (**a**) Experimental setup for focal depth experiment. (**b**) Front view photograph of the layered metal object. (**c**) Imaging result of the layered metal object.

Finally, the proposed imaging scanner system was used in a real imaging experiment, as shown in Figure 12. The video of the real imaging experiment can be seen in Supplementary Video S1. A $50 \times 20$ mm$^2$ metal object (Figure 13a) was held by hand and covered with a piece of polyester fabric. The scanning frequency of the polygon mirror was 250 Hz, and the hidden metal object was moved by hand along the z-direction at a speed of around 500 mm/s. The acquisition time was approximately 0.05 s. The hand was in an unstable position when moving at high speed. In this case, the integrated OAP mirror can take advantage of its enhanced focal depth compared to the f-theta lens in Figure 2a. In order to simulate the situation in real applications, we used two different thicknesses of polyester fabric to cover the metal object. Figure 13b,c show the imaging results of the metal object under 0.05 and 2 mm thicknesses of polyester fabric, respectively. The metal object behind the polyester fabric can be distinguished clearly. Therefore, the proposed system has the potential to be used for the real-time imaging of moving objects, such as in the application of security and food inspection.

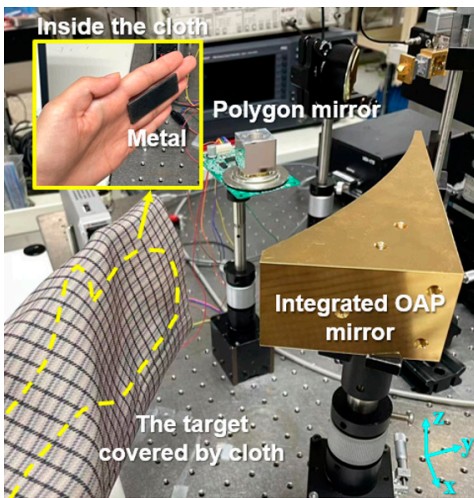

**Figure 12.** Photograph of the experimental setup for a real imaging experiment.

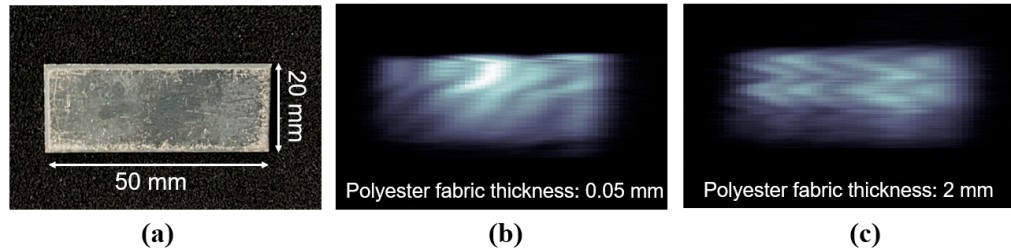

| (a) | (b) | (c) |

**Figure 13.** Imaging results of a real imaging experiment. (**a**) Photograph of the metal object. (**b**) Imaging result of the metal object under 0.05 mm thickness polyester fabric. (**c**) Imaging result of the metal object under 2 mm thickness polyester fabric.

## 4. Conclusions and Outlook

In conclusion, we introduced a fast polygon mirror to increase data acquisition speed and designed and tested an integrated OAP mirror to enhance the focal depth of our 600 GHz-band THz imaging scanner system. The experimental results of the polygon mirror show that a spatial resolution of approximately 2 mm can be obtained at the polygon mirror's scanning frequency of 250 Hz and a moving stage speed of 500 mm/s. In addition, the parameters of the integrated OAP mirror were mathematically calculated, and the performance was evaluated using both simulations and experiments. The experiments confirmed that the imaging scanner system has a spatial resolution of approximately 2 mm at a focal depth of 170 mm. Finally, the proposed system was successfully applied to image hand-held moving metal objects covered by two different thicknesses of polyester fabric as an example of a practical application.

In future work, the modulation transfer function (MTF) will be used to define the upper limit of the spatial resolution of the proposed system. Moreover, 3D imaging techniques should be combined with the proposed system to achieve real-time 3D imaging. Another potential task is to miniaturize the THz imaging system by introducing integration technologies [4]. A schematic diagram of the miniaturized THz imaging system and the integrated unit structure are shown in Figure 14a,b, respectively. A resonant tunneling diode (RTD) chip as a transmitter [33], an SBD chip as a receiver [34], and a beam splitter are integrated on a silicon wafer [23]. By introducing an integrated unit, our 600 GHz-band THz imaging scanner system can be miniaturized, which can increase its potential use in practical applications, such as security inspections. In addition, the performance of the proposed OAP mirror can be further enhanced, including a more accurate design methodology.

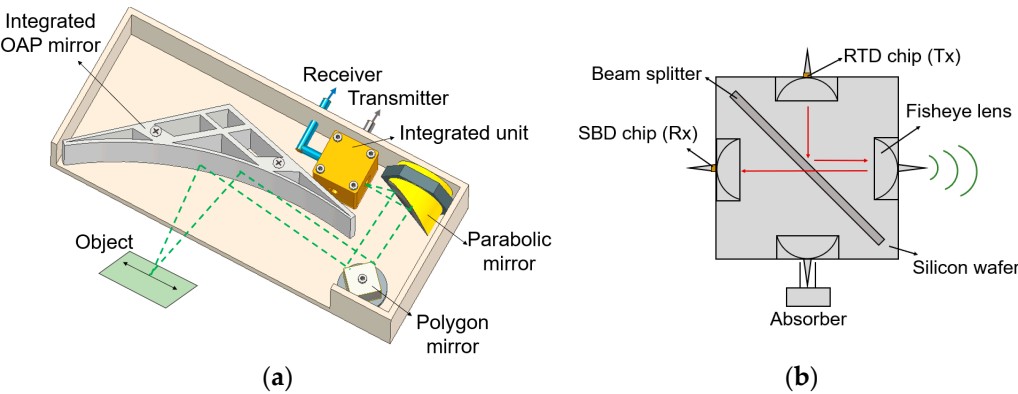

**Figure 14.** (**a**) Schematic diagram of the miniaturized THz imaging system. (**b**) Schematic diagram of integrated unit structure. (RTD: resonant tunneling diode, SBD: Schottky barrier diode.).

**Supplementary Materials:** The following supporting information can be downloaded at: https://www.mdpi.com/article/10.3390/photonics9120913/s1, Video S1: Video of the experimental setup for a real imaging experiment.

**Author Contributions:** Conceptualization, Y.W. and L.Y.; methodology, Y.W.; software, Y.W.; validation, Y.W., L.Y., M.T. and T.N.; formal analysis, Y.W.; investigation, M.T. and T.N.; resources, L.Y., M.T. and T.N.; data curation, Y.W.; writing—original draft preparation, Y.W.; writing—review and editing, L.Y., M.T. and T.N.; visualization, Y.W.; supervision, L.Y.; project administration, T.N.; funding acquisition, L.Y. and T.N. All authors have read and agreed to the published version of the manuscript.

**Funding:** This research was funded by the Japan Society for JSPS Grants-in-Aid for Scientific Research Grant-in-Aid for Early-Career Scientists (19K14995), in part by KAKENHI Japan (20H00253), and in part by the National Institute of Information and Communications Technology (NICT), Japan, commissioned research (02801).

**Institutional Review Board Statement:** Not applicable.

**Informed Consent Statement:** Not applicable.

**Data Availability Statement:** Data supporting the findings of this study are available from the corresponding author upon reasonable request.

**Acknowledgments:** The authors would express their special thanks to Masaya Nagai for his kind help in fabricating the proposed mirror and to JFE Shoji Electronics Corporation for their cooperation and support.

**Conflicts of Interest:** The authors declare no conflict of interest.

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
