# Peer review of "High-Speed 600 GHz-Band Terahertz Imaging Scanner System with Enhanced Focal Depth"

_photonics, doi:10.3390/photonics9120913_

Round 1

Reviewer 1 Report

This manuscript investigated to improve the short-focal depth of the existing system from a practical point of view using the astigmatism of the OAP mirror. I have a few major points.

1. First, in Section 2, authors showed that high-speed scanning is possible by realizing a system using an f-theta lens and a polygon mirror. You can physically rotate polygons quickly and move samples quickly. However, more importantly, the faster the scan, the shorter the time allotted for signal processing for one point. As a result, the signal processing time must be faster, which also increases the noise signal. It looks like we need more information in this regard. For example, time constant [s], sampling rate [data/sec], noise level change, etc ..

2. The author emphasized in Section 3 that OAP mirrors can be used to implement systems with long depth of focus. Using the system shown in paragraph 2, what if you compare graph data according to the distance of an object as shown in Figure 10(b), and furthermore, author said that the OAP mirror increases the focal depth but reduces the spatial resolution to some extent. In this regard, I'm wondering if it's possible to show a 2D image with respect to the sample in Figure 5' or the sample in Figure 10.

3. It seems a little difficult to claim the advantage of the long focal depth of the system using the OAP mirror only with the data shown in Fig. 12 using the proposed system. Since terahertz waves penetrate the polyester fabric well, rather than simply showing the difference depending on the thickness of several mm, for example, how about simultaneously scanning two or more samples at a distance of a few centimeters from the system inside a clothes?

4. How is the OAP mirror machined?

5. There is a unit typo on line 181. Change '180 m' to '180 mm'

Reviewer 2 Report

Review of the Manuscript

“High-Speed 600-GHz-Band Terahertz Imaging Scanner System…” by Yaheng Wang, Li Yi, Masayoshi Tonouchi and Tadao Nagatsuma.

 In this Manuscript was introduced an imaging scanner with polygon mirror to increase important for data acquisition speed, which was designed with an integrated off-axis parabolic mirror to enhance the depth field of this system. The subject considered there is actual from the point of view of designing of fast acquisition facilities in the THz spectral range for many practical applications.

As for me, this Manuscript worth to be published in “Photonics” with tiny corrections.

 Some small remarks:

-            P. 1, line 26. Perhaps, as it seems to me, it would be reasonable to Refs. [1-3] add or change some of them the following Refs., which are important for evaluation the state of the art of this subject, e.g.: D.M. Mittleman, Twenty years of THz imaging, Opt. Express, 26, 9417 (2018); G. Valusis, et.al., Road map for THz imaging, Sensors, 21, 4092 (2021).

-            P. 4, Fig. 2.a. For clearness, it would be better to shift a little the title “F-theta lens” aside of dashed line as the letter “l” coincide with dashed line.

-            P. 6, line 181. “…the radius is 180 m” - ?

-            P. 9, line 270. From my point of view, it would be better to present, at least, one picture from Ref. [32] as this Ref. is not easily acceptable for plenty of Readers.

-            P. 10, Line 282. Perhaps the Authors should add some other feasible applications, e.g., food-control lines, for which such system really can be suitable, though such kind of system seems, is not cost-effective.

-            P. 12, Ref. [22]. It seems to me it is better to mention the update data of this Ref., year, and correct the Author name - T. O’Haver.

-            Evaluated resolution of 2 mm (by metal hole in the metal erasing shield) for radiation frequency 600 GHz is more or less acceptable for many real-time imaging applications but it would be not bad, maybe in the next research, define the upper limit value of the system resolution using the Modulation Transfer Function (MTF).

Reviewer 3 Report

The paper titled "High-Speed 600-GHz-Band Terahertz Imaging Scanner System with Enhanced Focal Depth" introduced a fast polygon mirror to increase the data acquisition speed. It is designed and tested using an integrated off-axis parabolic mirror to enhance the focal depth of a 600-GHz-band THz imaging scanner system. The idea behind the system and the methodology are unique. The setup and the procedures are described in a clear way. The results of the paper are outstanding. Hence, I recommend its publication in this journal. Nevertheless, I have some comments that can further enhance the paper.
1- What's the effect of humidity at such a quite high frequency of 600 GHz? 2- Can the authors comment on the effect of the signal quality at this frequency on the focal length, if any, and the resolution? 3- If the sample is not flat, how that will affect the image? What do you suggest to do in such a scenario?"

Reviewer 4 Report

The manuscript “High-Speed 600-GHz-Band Terahertz Imaging Scanner System 2 with Enhanced Focal Depth” by Yaheng Wang and co-workers presented an innovative work by using an integrated OAP mirror to increase the focal depth and spatial resolution in THz imaging and showed potential application for homeland security. The authors presented a detail study by comparing with a well know method of using a f-theta lens.

I recommend the manuscript for publication, but some typo and clarification needed to be corrected:

Page 3, Line 95 looks like there is a misinterpretation of parameters between  ,  and . “where θ is the focal length of the f-theta lens, and h is half of the beam aperture, a, as 95 shown in Fig. 1(a).”

Many similar parameters are used for different equations but hold a different meaning, more subscripts should be used. Page 6, Line 175,  is used again but for off axis angle. Line 177 of the same page, “where r is the radius” would be good to add radius of the mirror unless it is the radius of the beam. Similarly on Page 7 Line 202,  is used again but for imaging area this time.

I would like to ask if the integrated OAP method could produce a better image in Figures 4 and 5 as it is not limited by the 600 GHz of the f-theta lens.

Likewise, can the f-theta method be compared against Figure 12 using the integrated OAP.

What is the limitation of the setup, if focal depth and spatial resolution is dependent on the OAP or the f-theta lens. What about scanning frequency, stage speed and acquisition rate as I see Ref 9 uses the same rotating polygon?

Round 2

Reviewer 1 Report

i think this manuscript deserves to be published in a journal. And thanks to the authors for their efforts.